# Mobile Computer Vision-Based Applications for Food Recognition and Volume and Calorific Estimation: A Systematic Review

**DOI:** 10.3390/healthcare11010059

**Published:** 2022-12-26

**Authors:** Lameck Mbangula Amugongo, Alexander Kriebitz, Auxane Boch, Christoph Lütge

**Affiliations:** Institute for Ethics in Artificial Intelligence, School of Social Sciences and Technology, Technical University of Munich, 80333 München, Germany

**Keywords:** computer vision, mobile applications, food recognition, volume estimation, nutritional monitoring

## Abstract

The growing awareness of the influence of “what we eat” on lifestyle and health has led to an increase in the use of embedded food analysis and recognition systems. These solutions aim to effectively monitor daily food consumption, and therefore provide dietary recommendations to enable and support lifestyle changes. Mobile applications, due to their high accessibility, are ideal for real-life food recognition, volume estimation and calorific estimation. In this study, we conducted a systematic review based on articles that proposed mobile computer vision-based solutions for food recognition, volume estimation and calorific estimation. In addition, we assessed the extent to which these applications provide explanations to aid the users to understand the related classification and/or predictions. Our results show that 90.9% of applications do not distinguish between food and non-food. Similarly, only one study that proposed a mobile computer vision-based application for dietary intake attempted to provide explanations of features that contribute towards classification. Mobile computer vision-based applications are attracting a lot of interest in healthcare. They have the potential to assist in the management of chronic illnesses such as diabetes, ensuring that patients eat healthily and reducing complications associated with unhealthy food. However, to improve trust, mobile computer vision-based applications in healthcare should provide explanations of how they derive their classifications or volume and calorific estimations.

## 1. Introduction

According to the European Regional Obesity Report 2022 by the World Health Organisation (WHO), about 59% of adults in Europe are obese or overweight [1]. Excess calorie intake, which is linked to unhealthy food consumption and nutritional imbalance, is one of the leading causes of obesity [2]. Conventionally, manual dietary assessment methods such as 24 h dietary recall (24HR) have been used, proving efficient means of assisting users to understand their dietary behaviour and allowing targeted interventions to address underlying health challenges [3]. However, 24HR requires the user to manually report their food consumption in the last 24 h period without the supervision of an experienced dietitian. As a result, the consumed food size reported reliant on the user’s visual assessment for estimation. This implies that the reported consumed portion could vary depending on the user’s judgement, which may lead to biased and inaccurate dietary information. To address the inaccuracy in dietary monitoring and assessment, semi-automated and automated systems have been previously proposed in the literature [4,5]. Advances in artificial intelligence (AI) have enabled the development of many applications with the potential to change how people monitor their health [6]. In addition, the easy availability and widespread use of mobile devices such as smartphones with integrated high-quality cameras make these devices practical for eating habits analysis in real life.

This review examines peer-reviewed studies covering mobile computer vision-based applications for dietary assessment. After a compressively systematic search of the literature, high-impact studies were reviewed. Thereafter, the techniques were compared to highlight the focus of the study, the type of dietary information provided and whether the algorithm provides any explanations to the user on factors that influence a particular prediction of said operation.

Though several reviews on food recognition and volume estimate have been published, most studies are focused on a particular research area. Ref. [7], for example, reviewed image-assisted dietary approaches, and another comprehensive study, Ref. [8], focused on sensor technologies for food intake monitoring. Boushey et al. [9] reviewed an extensive number of techniques that use image-assisted and image-based approaches for dietary assessment. In addition, the authors discussed the benefits and challenges of the different approaches reviewed. In addition, Ref. [10] explored computational models, mathematics and techniques applied in image-based dietary assessment. Finally, a recent survey study discussed algorithms developed for automatic food recognition and volume estimates for dietary estimation [11], which highlighted the need for transparency, using explainable AI as a gap to be filled by future studies.

Today, the majority of image-based diet assessment systems rely on machine learning (ML) approaches, and more specifically deep learning (DL), to recognise food types, estimate the volume and predict the nutritional value of a given dish [12]. A major issue arising from the use of DL-based approaches is the “black box” aspect of the systems, as they do not provide insights into factors influencing the decision making of the algorithm due to their inherent opaqueness. For a computer vision-based diet recommendation system to be adopted in contexts such as healthcare to, for instance, support diet planning for patients with chronic illnesses such as diabetes, dietitians and clinicians need to understand to trust the decisions made by the AI-powered system [13]. Therefore, to improve the trustworthiness of AI-based solutions in healthcare, explainability and interpretability is necessary.

In this study, we extensively review existing mobile-based applications used to detect and recognise food to measure the amount of nutritional intake, focussing on the underlying algorithms and approaches applied for accurate volume and calorie estimation. In the context of the investigation proposed here, we define the "mobile” part of the term as solutions for edge devices (smartphones, raspberry/Arduino). In addition, we assess the extent to which these applications explain to the user factors that influence the decision making of the model. Several authors have defined explainability differently in the literature [14,15]. In the context of our study, we simply define explainability as the ability of the model to explain the internal working mechanism to humans, i.e., how the algorithm makes decisions. The main contributions of this study are as follows:To the best of our knowledge, this is the first systematic review that focuses on mobile computer vision-based algorithms for food recognition, volume estimation and dietary assessment to determine the extent to which existing computer vision-based applications provide explanations to help the users understand how the algorithms make decisions.The analysis proposed provides a critical comparison among mobile-based automatic food recognition and nutritional-value-estimation techniques.This study analyses gaps and proposes possible solutions to create trustworthy image-based food recognition and calorie estimation applications for nutritional monitoring.

The remainder of this study is organised as follows: the methods section describes the approach used to determine the eligibility of published articles used in this study; the results section discusses the findings of the analysis; finally, the discussion section lays out the conclusions stemming from the interpretation of the findings.

## 2. Materials and Methods

A systematic review of the literature uses an explicit and reproducible research approach to methodically search the existing published work and find results of multiple studies on a similar topic. This process aims to summarise the state of the art and to answer fundamental research questions on particular and defined issues [16,17].

In this section, we outline the detailed methodology used to carry out the systematic review. The systematic review was conducted following the PRISMA (Preferred Reporting Items for Systematic Reviews and Meta-Analyses) [18]. In this study, all manuscripts included met the criteria, as illustrated in Table 1.

The eligibility criteria that were used to select the articles were as follows:Only articles available in English.Only articles published between January 2010 and October 2022.Only papers that discuss computer vision systems on mobile phones for food recognition, volume estimation and calorie estimation.

The advances in computing technology on mobile phones have proliferated the scientific research interest in the use of mobiles over the last 12 years. In this study, we included articles published between 2010 and 2022. Figure 1 shows the distribution of studies retrieved from PubMed, IEEE Xplore and Scopus grouped by the year of publication.


**Exclusion criteria**


We excluded articles or papers that met the following: (1)Short conference papers;(2)Review articles;(3)Full-text not available.

Additionally, studies that did not meet the criteria summarised in Table 1 were excluded from this systematic review.

### 2.1. Search Methods

The following databases were used, IEEE Xplore, PubMed and Scopus, to identify and collect articles related to mobile-based computer vision for nutritional monitoring. The selection was performed based on relevance to the domains of interest and scope. The fields considered in the search query were limited to the titles and abstracts of the papers. Several keywords were used, combining them using Boolean operators (AND, OR and NOT) to cross-examine the scientific databases. As an example, the search keywords used for PubMed, IEEE Xplore and Scopus are provided in Table 2.

### 2.2. Selection of Studies

After retrieving the articles from the search databases, we used Mendeley reference manager software to create a database of references, remove duplicates and manage the references. As highlighted in the inclusion criteria, articles were selected based on a three-step process:Assessment of the title;Assessment of the abstract;Assessment of full article.

The three-step process is recommended by [19], with the aim of refining the database by removing irrelevant papers and ensuring that only papers that meet the eligibility criteria are reviewed. The full process used for selection, including screening and determining eligibility and inclusion, is illustrated in Figure 2.

### 2.3. Data Extraction

For each of the chosen studies, the following information was included: (1) focus of study; (2) dataset; (3) method; (4) result; (5) whether it provides diet information; (6) explainability. In this study, explainability was assessed in terms of the system’s ability to provide the rationale behind a particular prediction—for example, why the algorithm recognised a particular dish to contain carbohydrates or why the algorithm estimated a given dish to contain a certain number of calories. However, we did not assess the quality of explanations given. The results were critically discussed, illustrating the state of the art of computer vision-based mobile applications for food recognition and volume estimation. Finally, we performed statistical analyses on the survey studies.

## 3. Results

### 3.1. Study Selection

We selected a total of 22 studies published between 2010 and 2021. These studies were selected from 393 articles retrieved from Scopus, PubMed and IEEE Xplore, after multiple elimination steps. The first step eliminated 25 studies because of duplication. The second step, screening of abstracts and titles, excluded 244 studies. Two more studies were excluded because of the lack of access to the full text. Finally, this systemic review concluded with 22 studies that aimed at using mobile applications for food recognition, volume estimation and calorific estimation. A schematic workflow for food detection, volume estimation and calorific estimation can be seen in Figure 3.

### 3.2. Food Recognition

Food recognition applications aim to identify and recognise food on a given dish with precision. Therefore, accurate, robust and trustworthy food classification algorithms are important to reach the expected level of outcome. As highlighted in a recent review by Tahir and Loo [11], several approaches have been used to classify food categories using image recognition. These approaches range from traditionally using manually crafted features to using more complex deep-learning-based characteristics. However, only a few of the proposed food image recognition systems have been tested in a real-life environment using mobile devices such as smartphones. 

Earlier mobile-based classifiers in the domain of food recognition include support vector machine (SVM), K-nearest neighbour (KNN) and multi-kernel learning. These algorithms have been preferred due to their higher performance when compared to other classification approaches. In their study, Kawano et al. [20] proposed mobile food recognition using manual features such as colour and texture to train an SVM. Similarly, Zhang et al. [21] developed a smartphone-based application for food recognition called “Snap-n-Eat” using a linear SVM classifier. The authors performed hierarchical segmentation to partition the food images into different regions. After they extracted low-level features from each region, these features were used by the classifier to determine the food category. Their SVM-based classifier achieved accuracy above 85% when identifying 15 categories of food. However, their system only categorised 15 types of food. 

Silva et al. [22] proposed an interactive mobile application for automatic food detection using a quadratic SVM on an expanded database containing 60 food classes. Their SVM classifier using colour, histogram of oriented gradients (HOG), modified local binary patterns (LBP), Gabor and speeded up robust features (SURF) performed better compared to the standard model on a validation Food-101 dataset. In addition, using DL-based features, they achieved an overall performance of 87.2%. 

Today, DL-based approaches are increasingly being proposed for food recognition. DL-based methods are being preferred because of their ability to learn automatically important features to distinguish the different food classes. Commonly used DL-based techniques in food recognition include convolutional neural network (CNN) [23,24], Deep CNN [25], InceptionV3 [26] and ensemble algorithms [27].

Temdee and Uttama [28] applied transfer learning to train a CNN model on a dataset of 2500 images with 40 categories. They reported a testing accuracy of 75.2%. Tiankaew et al. [29] proposed a food photography application for smartphones using transfer learning to adapt their deep CNN model to learn from a dataset of 7632 images of 13 kinds of Thai food collected from the Internet. They achieved a test accuracy of 82%. Similarly, Ref. [30] developed a deep CNN model for Korean-food image detection and recognition for use on mobile devices. Their model achieved a test accuracy of 91.3% in detecting 23 categories of Korean food. The aforementioned studies are limited by the few food categories they can detect.

Food recognition can be a tedious task that requires a large, diverse dataset to achieve good accuracy in recognising different types of food. Mezgec and Seljak [25] proposed “NutriNet” fine-tuned on a dataset composed of 225,953 images and presenting 520 categories of food and drinks. The model achieved a top-5 accuracy of 55% on real images taken with a mobile phone. Despite the generally good performance of their model, the authors did not perform segmentation, implying that irrelevant items in the image create an image recognition challenge. On the other hand, Freitas, Cordeiro and Macario [31] developed a segmentation approach using a region-based convolutional neural network (RCNN) to classify Brazilian food types. With their CNN-based segmentation model integrated into a mobile application, their segmentation analysis achieved an intersection over the union (IoU) accuracy of 0.70. Table 3 summarises food recognition and classification methods used in mobile applications.

Despite DL-based approaches outperforming traditional food recognition methods, DL methods are often applied to large unlabeled datasets because data annotation of large databases remains a challenge. Thus, applying DL techniques to unlabeled data can be challenging and less effective. To overcome this challenge, methods exploring both mid-level-based and deep CNN techniques have been proposed [32]. However, such an approach will usually employ many different features and extremely deep CNN architectures, which can be computationally expensive. Thus, it is not suitable for usage on mobile phones. Further research focusing on developing lightweight and computationally efficient DL models will enable deploying mobile-based deep CNN applications.

**Table 3 healthcare-11-00059-t003:** Summary of food recognition techniques used by existing computer vision-based mobile applications.

Author	Focus	App Name	Dataset (Categories)	Algorithm	Features	Accuracy (Top 5)	Distinguish Food/Non-food	Explainability	Mobile Platform
Kawano and Yanai [33] 2013	Food recognition	FoodCam	6781 images (50)	Linear SVM	histogram, Bag-of-SURF.	(81.55%)	No	No	Android
Zhang et al. [21] 2015	Food recognition	Snap-n-Eat	(15)	Linear SVM	Colour, HOG, SIFT, gradient.	85%	No	No	Android
Mezgec and Seljak [25] 2017	Food and drink recognition	-	225,953 images (520)	Deep CNN adapted from AlexNet	CNN-based features	86.39% (55%)	No	No	Mobile-web
Silva et al. [22] 2018	Food recognition	-	Extended Food-101	Quadratic SVM	Gabor and SURF features.	-	No	No	Android
Temdee and Uttama [28] 2017	Food recognition	-	2500 images (40)	CNN	Filter based on three RGB colour channels.	75.2%	No	No	Mobile-web
Termritthikun, Muneesawan, and Kanprachar [34] 2017	Food recognition	-	THFOOD-50	CNN	CNN-based features	69.8% (92.3%)	No	No	Android
Tiankaew et al. [29] 2018	Food recognition	Calpal	7632 images (13)	CNN and adapting VGG19	CNN-based features	82%	No	No	Cross-platform (Android and iOS)
Qayyum and Şah [23] 2018	Food image recognition	-	5000 images	Modified CNN	CNN-based feature	86.97% (97.42%)	No	No	iOS
Sahoo et al. [35] 2019	Food image recognition	FoodAi	FoodAI-756	Transfer learning. CNN	CNN-based feature	80.09%	No	No	Mobile-web
Park et al. [30] 2019	Food recognition	-	92,000 images (23)	DCNN	CNN-based features	91.3%	No	No	Mobile-web
Kayikci, Basol and Dörter [36] 2019	Food classification	Türk Mutfağı	Food24	CNN	CNN-based features	93%	No	No	iOS
Freitas, Corddeiro and Macario [31] 2020	Food segmentation and classification	MyFood	1250 images (9)	Mask RCNN	CNN-based features	IoU = 0.70	No	No	Cross-platform (Android and iOS)
Cornejo et al. [26] 2021	Food recognition	NutriCAM	3600 (36)	CNN	CNN-based features	85%	No	No	Cross-platform (Android and iOS)
Tahir and Loo 2021 [27]	Food image analysis	-	Food/Non-Food, Food101, UECFood100, UECFood256, Malaysian Food.	MobileNetV3	CNN-based features with fine-tuning.	Food/Non-Food: 99.12%. Food101: 80.80% UECFOOD100: 80.40% UECFOOD256: 68.50% MalaysianFood: 71.2%	Yes	Yes	Android

CNN = convolutional neural network; DL = deep learning; DCNN = deep convolution neural network; FCN = fully convolutional neural network; IoU = intersection over union; RCNN = region-based convolutional neural network; SDG = stochastic gradient descent; SURF = speeded up robust features; SVM = support vector machine.

### 3.3. Volume Estimate

After food recognition itself, the next challenge for automatic food analysis is to estimate the volume of food on a given plate. The first challenge for this task is the precise identification of each type of food present in a dish, as individual materials might be cooked or prepared differently from one meal to the other (i.e., fried, baked, or cooked). Additionally, the quality of a picture might vary for different mobile devices, which could affect the accuracy of volume estimation of food on a given image. Therefore, algorithms need to perform enhancements on images or avoid making volume estimations if the image quality does not yield accurate results. Methods for volume estimation employed in mobile applications are summarised in Table 4.

In the literature, several techniques have been proposed for volume estimation, ranging from simple techniques such as an approach based on pixel counting [21] to more complex three-dimensional (3D) model segmentation [24,37]. In their application called “Snap-n-Eat”, Zhang et al. [21] used pixel counting to estimate the portion size of each segmented food section on a given plate. Once the portion size is computed, the authors could estimate the calories and nutritional facts present on the plate. They found that the pixel-counting approach was simple and gave a good estimation of portion size. However, they assumed predefined calorific and nutritional value per food category. This assumption may not be true; for example, an ounce of baked potato chips can have 14% fewer calories, 50% less fat and less saturated fat than fried potato chips [38].

Rhyner et al. [37] used a 3D model and segmentation results of each food item on a plate to compute carbohydrate content on a given plate. Their mobile-based application, “GoCARB”, provided more accurate carbohydrate estimations when compared to traditional methods in a similar cohort of type 1 diabetes participants. However, this approach has some limitations, such as identifying complex meals with multiple ingredients or meals covered by sauces. In another study, food volume was estimated using depth map fusion from smartphone images taken from different angles [24]. The volume estimate was derived from a 3D model of the food object. The results show an accurate and reliable food volume estimation, though with a slight overestimation of 0–10% depending on the shape of the object. Generally, 3D modelling for food volume estimation approaches gives superior estimations compared to methods building on single-view images.

Vision-based dietary assessment applications have demonstrated reasonable accuracy in estimating food portions. Several key challenges such as view occlusion and scale ambiguity have been reported in the literature. Moreover, the proposed approaches require users to take multiple images of a given dish from different viewing angles. Acquiring multiple images before eating can be tedious and may not be practical for long-term dietary monitoring. As a solution to view occlusion and scale ambiguity, depth sensing techniques have been proposed for volume estimation, combining depth sensing and AI capabilities [39,40]. The real-time 3D reconstruction with deep learning synthesis demonstrated better volume estimation compared to previous approaches. However, these algorithms are yet to be fully tested in a real-world setting using a mobile phone application.

**Table 4 healthcare-11-00059-t004:** Summary of food volume estimation methods used in mobile applications.

Author	Focus	Dataset (Categories)	Method	Features	Result (Error)	Explainability	Application Name
Zhu et al. [41] 2010	Volume estimation	3000 images	Step 1: camera calibration. Step 2: 3D volume reconstruction. (multi-view)	Fiducial markers	Error: 1%	No	-
Zhang et al. [21] 2015	Calorie estimation	(15)	Counting pixels in each segmented item. Additionally, using the depth of the image. (single-view)	SIFT features and HOG features.	85%	No	Snap-n-Eat
Akpa et al. [42] 2016	Volume estimation	119 images	Image processing with chopstick		Error: 6.8%	No	-
Rhyner et al. [37] 2016	Carbohydrate estimation	19 adults (n = 60 dishes; 6 dishes a day)	3D model and segmentation. (multi-view)	Colour and texture.	(Error: 18.7%)	No	GoCarb
Okamoto and Yani [43] 2016	Calorie estimation	60 test images (20)	Quadratic curve estimation. (single-view)	2D size of food	Error: 21.3%	No	
Silva et al. [22] 2018	Estimate weight and calories	Food-101	Estimated food volume from segmented food. With fingers as reference. (single-view)	CNN based features	(error +/− 5% and 8% of ground truth)	No	-
Tiankaew et al. [29] 2018	Calorie estimation	(13)	Compute calories. (single-view)	User information and calorie table.		No	Calpal
Gao et al. [44]	Volume estimation	SUEC Food	Multi-task CNN (single-view)	Deep CNN	Error: Chicken: 2.7% Fried pork: 12.3% Congee: −0.27%	No	MUSEFood
Sowah et al. [45] 2020	Calorie estimation and recommendations	300 (25)	Use Harris Benedict’s equation to determine calorie requirements. (single-view)	Patient data		No	-
Tomescu [24] 2020	Volume estimation	80,000 (382)	CNN EfficientNet (Multi-view)	Depth maps, shape.	10% volume overestimation.	No	-
Herzig et al. [46] 2020	Volume estimation	48 meals (128 items)	CNN for segmentation (single-view)	Depth sensing	Absolute error (SD): 35.1 g (42.8 g; 14% [12.2%])	No	-

2D = two-dimensional; 3D = three-dimensional; CNN = convolution neural network; DL = deep learning; FCN = fully convolutional neural network; HOG = histogram of oriented gradients (HOG); IoU = intersection over union; SIFT = scale-invariant feature transform; T1D = type 1 diabetes; T2D = type 2 diabetes; USDA = US Department of Agriculture.

### 3.4. Strengths and Weakness of Computer Vision Applications for Dietary Assessment

In this study, we summarised computer vision-based mobile applications used for food recognition, nutritional volume estimates and dietary assessment. Several mobile computer vision-based applications have been proposed in the literature for food recognition of Middle Eastern [36], European [24,37], American and Asian food [29,30]. These applications learn how to recognise food from images using existing large food databases, such as UNIMB 2016, PFID, Food-101, UECFOOD and Vireo-Food. However, most of these systems do not filter out non-food images except for one recent application [27].

Mobile devices are resource-constrained environments, having limited processing power and short battery lives. The emergence of mobile devices with sufficient resources to run ML models locally is promising. Such technology allows one to keep user data on the device, thereby reducing privacy concerns and server load. However, the majority of the existing mobile solutions proposed for food detection and volume estimation perform ML on servers, which could lead to possible unethical consequences; indeed, a such solution requires user (personal) data to be transmitted to external servers. Therefore, mobile health applications need to perform operations on the users’ devices to reduce ethical risks relating to personal data. Some studies have demonstrated the promising performance of lightweight neural networks, which can be used in smartphone applications [47,48]. Edge ML is possible by reducing the model size (number of parameters). 

Existing computer vision-based applications are useful in helping users monitor dietary intake, provide quick results and offering responses to users at scale. Existing applications use different methods and techniques. As a result, food recognition and volume estimation performance differ in terms of accuracy. Hence, there is a need to standardise the applied methods to improve reproducibility. Additionally, there are no existing ethical guidelines to describe errors, measure bias or address other ethical concerns. Therefore, there is a need for ethical principles to be incorporated into the design and development of computer vision-based applications for food recognition and volume estimation.

### 3.5. Explainability

Out of the solutions proposed in the literature, only one study proposed a user-centred AI framework to increase trustworthiness in the ingredient detection algorithm [27]. In their study, the authors of [27] used SHAP (Shapley additive explanations) to highlight regions in a dish that contribute positively to the model prediction. SHAP applies a game theory approach to explain how the ML model makes predictions by highlighting features that most influence model predictions [49]. SHAP values have desired mathematical properties as a solution to game theory problems [50]. However, SHAP values are limited in that they ignore causal structures in the data [51]. Additionally, SHAP values are not a solution to human-centric explanations. Human-centric explanations can be achieved by linking explanations to domain knowledge [52]. Finally, developing explainable and interpretable mobile health applications will improve trust levels and increase the adoption of such applications in a real-world environment.

### 3.6. Statistical Analyses

The pie chart in Figure 4 represents the distribution of food datasets of the studies surveyed grouped by the country from where the food dataset came from. Three studies used more than one dataset from different countries, we categorised these studies as using a generic dataset. We summarised the surveyed studies in two groups: (1) studies that distinguished between food and non-food and (2) studies that did not. As shown in Figure 5, only two studies distinguished between food and non-food. Moreover, as highlighted in Section 3.5, only one study attempted to provide explanations on how the model makes decisions to the end users to improve trust. The pie chart in Figure 6 shows the percentages of studies of different types, attempting to explain classification or prediction results.

## 4. Discussion

Here we discuss the results and how they can be interpreted from the perspective of previous studies and the working hypotheses. The findings and their implications should be discussed in the broadest context possible. Future research directions may also be highlighted.

In this study, we provided a systematic review of mobile computer vision-based approaches for nutritional monitoring. Our study focused on food classification methods using food recognition and nutritional volume estimation. Additionally, the review explored and compared the extent to which the proposed algorithms provide explanations to end users on the outputs of ML models. Moreover, due to its systematic approach, our study is reproducible. To the best of our knowledge, this is the first systematic review focusing on mobile computer vision-based applications for food image recognition, volume estimation and deriving nutritional value.

Previously, existing computer vision-based algorithms for dietary assessment have been reviewed. For example, Min et al. [53] examined emerging methods, concepts and tasks in food computing. In another study, Subhi et al. [54] presented an outline of methodologies used for automatic dietary assessment, including their performance, feasibility and challenges. In a recent study, Tahir and Loo [11] conducted a comprehensive survey to scrutinise traditional and deep visual methods for feature extraction and classification in food recognition. Unlike existing surveys, our study is the first systematic review focusing on mobile computer vision-based applications. Lastly, we reported whether existing mobile computer vision-based applications provide explanations regarding the algorithms’ predictions to the user.

### 4.1. Findings

The majority of the studies in the literature proposing mobile computer vision-based applications for food recognition do not filter non-food images. Distinguishing between non-food and food from images is an essential first step for any food recognition system. Several studies have reported better performance by employing transfer learning using pre-trained models and fine-tuning in this regard [23,25,29,55]. The majority of mobile applications for food recognition from images have neglected beverages. We only found one mobile application that recognises drink images. Drinks, especially alcoholic beverages, can have a negative impact on individuals’ health. Thus, mobile applications for dietary assessment should include drinks. Recognising drinks will be challenging, given the fact that drinks do not have a clear shape and are often occluded by their containers, and ingredients are often blended in the drink [56]. Thus, accurately recognising the nutritional content of a drink based solely on image seems arduous and ambitious. 

The majority of mobile applications for food volume estimates use a single-view method for food volume estimation. Though the single-view methods are friendlier for the users compared to multi-view approaches, they can be inaccurate given the fact that food is a 3D object. Therefore, multiple images are needed for accurate food estimation. In addition, standardised guidelines are needed to ensure robustness and consistency in food volume estimation algorithms. 

### 4.2. Challenges and Outlook

Advanced approaches such as deep learning-based methodologies have increased the performance and classification accuracy of food image recognition. However, as highlighted in several studies, deep learning approaches are limited to one output per image [25]. This implies that not every item gets successfully recognised, for example, in images with multiple foods or drink items.

Although we are witnessing a growing number of datasets, including diverse food categories, those datasets are not yet inclusive. Out of all the mobile vision-based applications we reviewed, only one application used African food (Ghanaian food) [46], and only two used Latin America: one Brazilian food [31] and one Peruvian food [26]. In addition, we observed that studies having used large datasets obtained better classification accuracy. Hence, there is a need to build a more inclusive large dataset for food images, to be frequently updated, using real-life images through crowdsourcing. Food recognition and volume estimation are complex and challenging tasks, which will require algorithms to understand the textures, shapes and appearances of different foods to handle the different variations of how different dishes are prepared. To achieve this, datasets comprising images, ingredients and other contextual information are needed. Moreover, there is no predetermined number for the number of food dishes. Hence, food recognition algorithms need to be agile to adapt to the continuously evolving variations of dishes. Despite their limitations, continuous learning approaches should be explored to determine how continuous learning can be applied to food datasets without forgetting previously learned information.

In this study, we reviewed computer vision-based applications for food recognition, volume estimation and dietary assessment. Our study is limited to mobile vision-based applications only. We conclude that food recognition and volume estimation mobile applications are still in their infancy, and because of a lack of trust in the algorithms composing them, these applications are yet to become a reality.

Therefore, designers and developers of these applications need to ensure that they provide user-friendly explanations, linking food segments in a given dish to nutritional information. The explainability of mobile computer vision-based models can help users, dietitians and clinicians understand, and therefore, trust the volume estimation performed by these systems.

## 5. Conclusions

In this study, we explored a wide range of computer vision-based mobile applications developed to detect food from images and estimate the calories in a given dish. Additionally, we examined the extent to which these food recognition applications provide explanations to users. We found that the majority of solutions proposed in the studies we surveyed do not distinguish between food and non-food. Similarly, only one out of 22 computer vision-based applications surveyed in our study have attempted to provide explanations by highlighting features that contribute towards an increase or reduction of a particular prediction.

Unsupervised computer vision-based applications, learning from unlabelled image datasets, are showing promising potential to create generalisable models able to perform classification on mixed food items. One major issue remains: unsupervised models are opaque in how they learn and make decisions. Explainability is required to provide understanding to the users on how and why a decision has been made by the algorithm. Therefore, unpacking the “black box” needs to become a priority to build trust towards the applications. Trust is key to increasing the adoption of mobile health applications. To achieve this, future AI-driven mobile health studies should incorporate ethical principles such as transparency, explainability and privacy from the onset. In addition, our results suggest that developers of mobile health applications should apply ethics by design to ensure that the solutions they develop are trustworthy.

Finally, dietary assessment applications should integrate multi-modality data such as videos and data from wearable devices to provide additional information to complement information from images.

## Figures and Tables

**Figure 1 healthcare-11-00059-f001:**
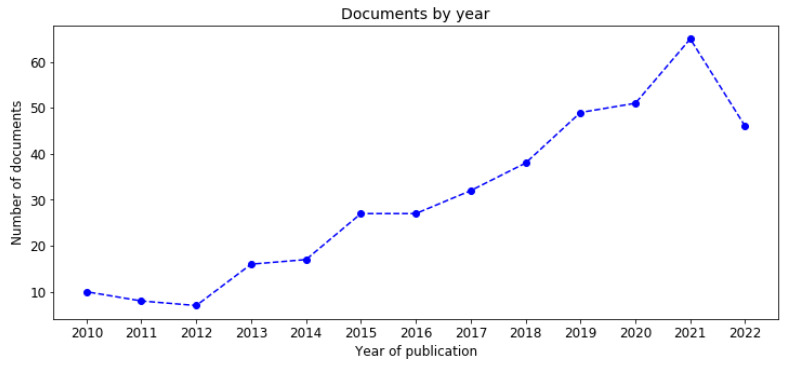
The distribution of documents is grouped by year of publication (PubMed, IEEE Xplore and Scopus).

**Figure 2 healthcare-11-00059-f002:**
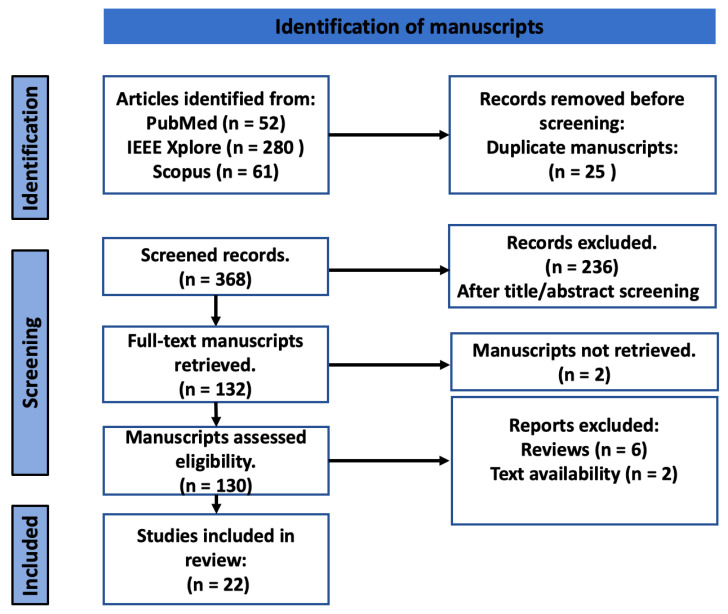
The diagram illustrates the PRISMA workflow.

**Figure 3 healthcare-11-00059-f003:**
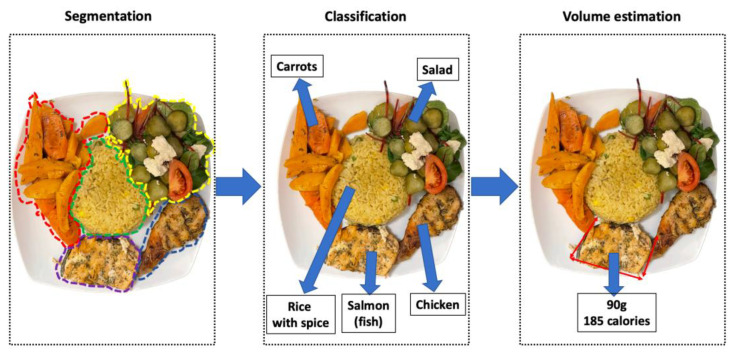
The schematic workflow shows the steps for image-based dietary assessment: segmentation, classification, volume and calorific estimation.

**Figure 4 healthcare-11-00059-f004:**
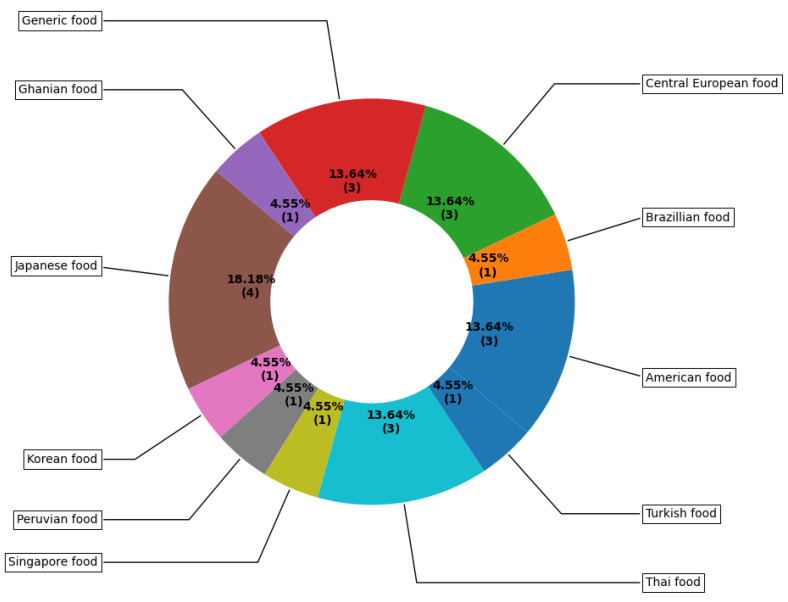
The percentages of datasets used in mobile computer vision-based applications for food recognition and volume estimation. Note: Generic food refers to when more than one multicultural dataset was used. The number in brackets refers to implemented vision-based applications that have used a particular dataset from surveyed studies.

**Figure 5 healthcare-11-00059-f005:**
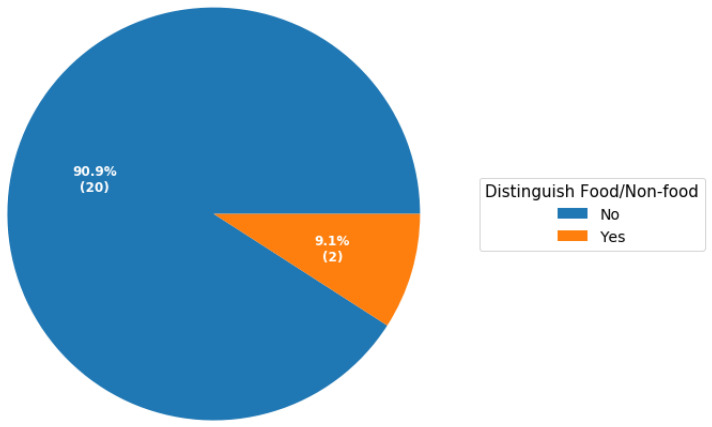
Percentages of mobile computer vision-based studies for food recognition and volume estimation that distinguish between food and non-food. The number of computer vision-based applications that distinguish food and non-food from studies survey is shown in brackets.

**Figure 6 healthcare-11-00059-f006:**
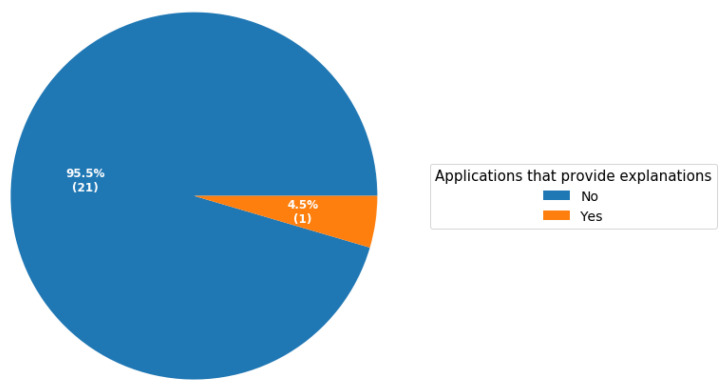
Percentage of mobile computer vision-based applications explain to users how the model makes predictions. The number in brackets indicates implemented computer vision-based applications that explain to users how the factors influence model prediction from the surveyed studies.

**Table 1 healthcare-11-00059-t001:** Summary of the inclusion criteria.

Criteria	Definition
Language of manuscript:	English
Years of publication:	2010–2022
Fields:	Artificial intelligence/computer vision Medicine/nutritional
The type of solutions considered:	Computer vision Healthy eating Nutritional estimate Food recognition Volume estimation
Types of device(s):	Mobile applications

**Table 2 healthcare-11-00059-t002:** Queries used for the selected search databases.

Search Database	Search Keywords
PubMed	(Nutritional monitoring [Title/Abstract]) AND (computer vision [Title/Abstract]) AND (artificial intelligence [Title/Abstract]) AND (smartphone [Title/Abstract]) AND (mobile [Title/Abstract]) OR (food recognition [Title/Abstract]) OR (Food images recognition [Title/Abstract])
IEEE Xplore	(“Abstract”: Nutritional monitoring) AND (“Abstract”: computer vision) OR (“Abstract”: Food images recognition) OR (“Abstract”: food image recognition) AND (“Abstract”: artificial intelligence) AND (“Abstract”: smartphone) AND (“Abstract”: Mobile)
Scopus	TITLE-ABS-KEY(“food image recognition” OR “food images recognition” OR “food volume estimation” OR “volume estimation” OR “nutritional monitoring”) AND TITLE-ABS-KEY-AUTH (“mobile device” OR “Mobile devices” OR “Smartphone” OR “Edge device”)

## Data Availability

Not applicable.

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
