# Peer review of "Mobile Computer Vision-Based Applications for Food Recognition and Volume and Calorific Estimation: A Systematic Review"

_healthcare, 2022, doi:10.3390/healthcare11010059_

Round 1

Reviewer 1 Report

The main research of this article is to present a systematic review of articles on mobile computer vision based food recognition, volume and calorie estimation applicationsdoing a relatively complete and clear work, which can be considered for acceptance, but needs the following modifications.

1.The article lacks sufficient analysis and discussion on the advantages and disadvantages of different methods, and it is recommended to add this part.

2. The tables in the article are too redundant and look a bit confusing, so I recommend optimizing them.

3. The authors didn't clearly indicate the direction of this research area. I recommend refining this section.

Author Response

We would like to thank the reviewers for their feedback.

Reviewer 1

The main research of this article is to present a systematic review of articles on mobile computer vision-based food recognition, volume and calorie estimation applications doing relatively complete and clear work, which can be considered for acceptance but needs the following modifications.

  1. The article lacks sufficient analysis and discussion on the advantages and disadvantages of different methods, and it is recommended to add this part.

We thank the reviewer for pointing out this weakness in our work.

Line 224 added: Despite DL-based approaches outperforming traditional food recognition methods. DL methods are often applied to large unlabeled datasets because data annotation of large databases remains a challenge. So, applying DL techniques to unlabeled data can be challenging and less effective. To overcome this challenge, methods exploring both mid-level-based and deep CNN techniques have been proposed (Zheng, Zou and Wang, 2018).  However, such an approach will usually employ many different features and extremely deep CNN architectures, which can be computationally expensive. Thus, not suitable for usage on mobile phones. Further research focusing on developing lightweight and computationally efficient DL models will enable deploying mobile-based deep CNN applications.

Zheng, J., Zou, L. and Wang, Z.J. (2018), Mid-level deep Food Part mining for food image recognition. IET Comput. Vis., 12: 298-304. https://doi.org/10.1049/iet-cvi.2016.0335

In line 534 in the discussion, “Moreover, there is no predetermined number for the number of food dishes. Hence, food recognition algorithms need to be agile to adapt to the continuously evolving variation of dishes. Despite its limitations, continuous learning approaches should be explored to determine how continuous learning can be applied to food datasets without forgetting previously learned information.”

  1. The tables in the article are too redundant and look a bit confusing, so I recommend optimizing them.

We thank the reviewer for the suggestion. We tried to remove the redundancy as much as we could and keep information that we think will be useful to readers. Nevertheless, it will be helpful if our esteemed reviewer can be more specific on which table and what we need to optimize.

  1. The authors didn't indicate the direction of this research area. I recommend refining this section.

We thank the reviewer for pointing out this omission.

In line, 422 added, “Developing explainable and interpretable mobile health applications will improve trust level and increase the adoption of such applications in a real-world setting.”

Reviewer 2 Report

The manuscript can be recommended for publication after the elimination of the following remarks: Line 44 and page 72. Offers with similar content, duplicate each other, I advise you to rewrite.

Table 3. The "Mobile platform" column is not entirely clear. Smartphone is not a mobile platform.

React Native is a cross-platform application development framework for both Android and iOS.

Table 4. Add abbreviations for SIFT, HOG, and so on.

Figures 4-6. Add an explanation to the numbers in brackets and describe what they mean.

Line 322-325 is apparently taken from the template, please check.

In the Conclusions, you can add information for whom the results will be useful.

Please check References for compliance with the MDPI format.

Author Response

We would like to thank the reviewers for their feedback.

Reviewer 2

  1. The manuscript can be recommended for publication after the elimination of the following remarks: Line 44 and page 72. Offers with similar content, duplicate each other, I advise you to rewrite.

We thank the reviewer for pointing out this duplication.

In line 44, rephrased “In this study, we conduct a systematic review of mobile computer vision-based algorithms used in the field of food image recognition for volume estimation and dietary assessment”.

New sentence:

This review examines peer-reviewed studies covering mobile computer vision-based applications for dietary assessment.

  1. Table 3. The "Mobile platform" column is not entirely clear. The smartphone is not a mobile platform.

We apologise to the reviewer for the confusion.

We thank the reviewer for the suggestion.  We clarified the mobile platform column.

Table 3: Changed React-native to Cross-platform (Android and iOS).

  1. React Native is a cross-platform application development framework for both Android and iOS.

We the reviewer for the suggestion.

Table 3: Changed React-native to Cross-platform (Android and iOS).

  1. Table 4. Add abbreviations for SIFT, HOG, and so on.

We thank the reviewer for this suggestion.

In the footnote of Table 4. We added the following abbreviations: scale-invariant feature transform (SIFT), Histogram of oriented gradients (HOG), 2D Two-dimensional, and 3D Three-dimensional.

  1. Figures 4-6. Add an explanation to the numbers in brackets and describe what they mean.

We thank the reviewer for the suggestion.

To the caption of Figure 4 added: “The number in brackets refers to implemented vision-based applications that have used a particular dataset from surveyed studies”.

To the caption of Figure 5 added: “The number of computer vision-based applications that distinguish food and non-food from studies survey is shown brackets”.

The caption of Figure 6 added: "The number in brackets indicates implemented computer vision-based applications that explain to users how the factors that influence model prediction from surveyed studies”.

  1. Line 322-325 is taken from the template, please check.

We thank the reviewer for the suggestion. Line 322 – 325 was checked to ensure that it conforms to the template.

  1. In the Conclusions, you can add information for whom the results will be useful.

We thank the reviewer for the suggestion.

In lines, 567-569 of the Conclusion, added: In addition, our results suggest that developers of mobile health applications should apply ethics by design to ensure that the solutions they develop are trustworthy.

  1. Please check References for compliance with the MDPI format.

We thank the reviewer for the suggestion. References were checked to ensure that they conform to MDPI

Reviewer 3 Report

The manuscript reviews a recently developing field of vision-based dietary intake assessment. 

1) The topic is recent and relevant to healthcare, and a review is timely and suitable for practitioners to come up to speed in this area.

2) But some up-to-date research works, especially for volume estimation, are missing. Deep learning-based volume estimation has been proposed over recent 5 years but limited articles using deep learning are reviewed in this paper. It would be great if the authors can also include both leaning-based 3D reconstruction method and the volume estimation method via depth estimation.

Some related articles are shown as follows for your reference:

Lu, Y., Stathopoulou, T. and Mougiakakou, S., 2021, January. Partially supervised multi-task network for single-view dietary assessment. In 2020 25th International Conference on Pattern Recognition (ICPR) (pp. 8156-8163).

Lo, F.P.W., Sun, Y., Qiu, J. and Lo, B.P., 2019. Point2volume: A vision-based dietary assessment approach using view synthesis. IEEE Transactions on Industrial Informatics, 16(1), pp.577-586.

Author Response

We would like to thank the reviewers for their feedback.

The manuscript reviews a recently developing field of vision-based dietary intake assessment. 

1) The topic is recent and relevant to healthcare, and a review is timely and suitable for practitioners to come up to speed in this area.

We thank the reviewer for the feedback.

2) But some up-to-date research works, especially for volume estimation, are missing. Deep learning-based volume estimation has been proposed over recent 5 years but limited articles using deep learning are reviewed in this paper. It would be great if the authors can also include both the leaning-based 3D reconstruction method and the volume estimation method via depth estimation.

Some related articles are shown as follows for your reference:

Lu, Y., Stathopoulou, T. and Mougiakakou, S., 2021, January. Partially supervised multi-task network for single-view dietary assessment. In 2020 25th International Conference on Pattern Recognition (ICPR) (pp. 8156-8163).

Lo, F.P.W., Sun, Y., Qiu, J. and Lo, B.P., 2019. Point2volume: A vision-based dietary assessment approach using view synthesis. IEEE Transactions on Industrial Informatics, 16(1), pp.577-586.

We thank the reviewer for highlighting this important omission in our paper. In our survey, we found a lot of deep learning-based papers, but they have not been implemented for mobile applications. The focus of our study is limited to methods for dietary assessment proposed for mobile applications.

Added the following:

In lines 353 - 363: Vision-based dietary assessment applications have demonstrated reasonable accuracy in estimating food portions. Several key challenges such as view occlusion and scale ambiguity have been reported in the literature. Moreover, the proposed approaches require users to take multiple images of a given dish from different viewing angles. Acquiring multiple images before eating can be tedious and may not be practical for long-term dietary monitoring. As a solution to view occlusion and scale ambiguity, depth sensing techniques have been proposed for volume estimation, combining depth sensing and AI capabilities (Lo et al., 2019; Lu, Stathopoulou and Mougiakakou, 2021). The real-time 3-D reconstruction with deep learning view synthesis demonstrated better volume estimation compared to previous approaches. However, these algorithms are yet to be fully tested in the real-world setting using a mobile phone application.

In line 541 added: Our study is limited to mobile vision-based applications only.